# The Systematic Assessment of the Membrane-Stabilizing and Antioxidant Activities of Several Kazakhstani Plants in the Asteraceae Family

**DOI:** 10.3390/plants13010096

**Published:** 2023-12-28

**Authors:** Alibek Ydyrys, Gulzhan Zhamanbayeva, Nazgul Zhaparkulova, Arailym Aralbaeva, Gulnaz Askerbay, Zhanar Kenzheyeva, Gulmira Tussupbekova, Sayagul Syraiyl, Raushan Kaparbay, Maira Murzakhmetova

**Affiliations:** 1Biomedical Research Centre, Al-Farabi Kazakh National University, al-Farabi Av. 71, Almaty 050040, Kazakhstan; 2Department of Biophysics, Biomedicine and Neuroscience, Al-Farabi Kazakh National University, al-Farabi Av. 71, Almaty 050040, Kazakhstan; gulzhan.kaznu.kz@gmail.com (G.Z.); nazgul.zhaparkulova@kaznu.kz (N.Z.); gulnaz.askerbay@kaznu.kz (G.A.); gulmira.tussupekova@kaznu.kz (G.T.); maira.murzakhmetova@kaznu.kz (M.M.); 3Scientific Research Institute for Issues in Biology and Biotechnology, Al-Farabi Kazakh National University, Al-Farabi Ave. 71, Almaty 050040, Kazakhstan; 4Faculty of Medicine and Health Care, Al-Farabi Kazakh National University, al-Farabi Av. 71, Almaty 050040, Kazakhstan; arailym.aralbayeva@kaznu.kz

**Keywords:** *Asteraceae*, antioxidants, plant extracts, flavonoids, lipid peroxidation, tea beverages

## Abstract

The objective of our research was to examine the antioxidant and membrane-protective characteristics of a few medicinal plant extracts belonging to the *Asteracea* family, along with their flavonoid and polyphenolic content, in order to identify strategies for enhancing beverage composition and boosting the antioxidant capacity of green and black tea. The activity of aqueous-ethanolic extracts from the dried parts of plants, such as *Arictum tomentosum* Mill., *Ghnapilum kasachstanicum* Kirp. & Kuprian. ex Kirp., *Artemisia schrenkiana* Ledeb., *A. rutifolia* Steph. ex Spreng., *A. cina* O.Berg, and *A. vulgaris* L., were examined using a model of Wistar rats. Thiobarbituric acid-reacting substances (TBARS), a marker of malondialdehyde concentration, were used to measure the amount of lipid peroxidation (LPO) in liver microsomes. Considering the outcomes, the extracts from *A. tomentosum*, *G. kasachstanicum*, and *A. vulgaris* exhibit the strongest membrane-stabilizing action among those examined. At a concentration of 5 g/mL, the extracts of these plants demonstrated a significant anti-hemolitic impact, whereas the remaining extracts displayed a similar effect at doses above 10 g/mL. Accordingly, among the extracts studied, the *A. tomentosum*, *G. kasachstanicum*, *A. schrenkiana*, *A. rutifolia*, *A. cina*, and *A. vulgaris* extracts have significant antioxidant properties. The integrated antioxidant and antihemolytic qualities of *A. tomentosum* and green tea extracts were comparable to those of the individual plant extracts. When the extracts of *A. schrenkiana* and green tea were combined, similar outcomes were seen, suggesting that there was no appreciable synergistic interaction.

## 1. Introduction

The Asteraceae family, commonly known as the aster or sunflower family, comprises a diverse array of plants that have been traditionally used for their medicinal and therapeutic properties across various cultures worldwide. Over the years, numerous phytochemical constituents from Asteraceae family plants have been identified and studied for their potential health benefits. Among the various bioactive compounds found in these plants, antioxidants and membrane-stabilizing agents have garnered significant attention due to their crucial roles in combating oxidative stress and maintaining cellular integrity [1].

Numerous chronic diseases are linked to oxidative stress, which is caused by an imbalance between the body’s capacity to neutralize reactive oxygen species (ROS) and their generation, including cardiovascular disorders, neurodegenerative conditions, and cancer. Consequently, the search for natural sources of antioxidants has become a subject of intense research interest. Simultaneously, the membrane-stabilizing properties of bioactive compounds have gained prominence for their ability to protect cell membranes from damage, thereby contributing to cellular health [2,3].

The genus Artemisia, a diverse group of plants belonging to the Asteraceae family, has a rich history of traditional use in various cultures for its medicinal properties. Several species within the Artemisia genus have been recognized for their pharmacological potential, with a particular focus on their antioxidant and membrane-stabilizing properties [4,5]. Antioxidants play a pivotal role in counteracting the harmful effects of oxidative stress, while membrane stabilizers are crucial for maintaining the integrity and function of cell membranes. In recent years, systematic investigations into the bioactive compounds of Artemisia species and their potential health benefits have gained momentum, making them a subject of significant scientific interest [6,7].

*A. vulgaris* (Common Mugwort) is another species within the Artemisia genus with a history of traditional use. It contains a variety of bioactive compounds, such as flavonoids, coumarins, and essential oils, which may confer antioxidant properties. The membrane-stabilizing potential of A. vulgaris extracts has also been explored in some studies, making it a relevant candidate for systematic analysis [8,9].

These are just a few examples of Artemisia species that have been investigated for their antioxidant and membrane-stabilizing properties. Systematic analysis of these and other Artemisia species could provide valuable insights into their potential therapeutic applications in mitigating oxidative stress and maintaining cellular integrity [10,11].

While there are several species of Arctium within the Asteraceae family, one of the most well-known and extensively studied species is *A. lappa*, commonly known as Greater Burdock. It is recognized for its traditional uses and potential medicinal properties. Here are some insights into its antioxidant and membrane-stabilizing properties [12]. *A. lappa* and *A. tomentosum* contain various bioactive compounds, including polyphenols, flavonoids, and lignans, which exhibit antioxidant properties. These antioxidants help combat oxidative stress by neutralizing reactive oxygen species (ROS) and free radicals, thereby protecting cells and tissues from damage [13].

The genus *Gnaphalium*, commonly known as cudweeds, comprises a group of plants within the Asteraceae family. While they may not be as well-known as some other genera, they have been traditionally used for various purposes, including potential medicinal applications. It’s important to note that research on the antioxidant and membrane-stabilizing properties of Gnaphalium species may not be as extensive as that of some other genera within the Asteraceae family. However, systematic analysis and further investigations could shed light on their potential therapeutic applications in combating oxidative stress and supporting cellular membrane health [14].

One of the major categories of food goods is tea beverages [15]. Because they contain beneficial compounds from fruits and berries, they can remedy some vitamin, microelement, and other nutritional deficits. Black or green tea may be the main ingredient in tea beverages, or they may just contain herbs with potential health benefits [16].

The *Asteraceae* family, also known as the aster or sunflower family, is one of the largest and most diverse plant families, containing many species with potential medicinal and therapeutic properties. These plants often contain a wide range of secondary metabolites, including polyphenols, flavonoids, terpenoids, and other compounds that contribute to their antioxidant activity. This investigation holds the promise of paving the way for the development of novel therapeutic agents and functional food supplements, contributing to the ever-expanding realm of natural medicine and wellness [17].

## 2. Materials and Methods

### 2.1. Laboratory Animals

The basic housing setup for male Wistar rats (220–230 g) included a light-and-dark cycle as well as free access to food and water. The livers and blood were extracted as stated below. The Kazakh Academy of Nutrition’s local ethics commission in Almaty, Kazakhstan, coordinated and approved the experiments (control No. 03/145 of 15 September 2015). All studies were conducted in accordance with institutional and national regulations that follow the ARRIVE guidelines and the directive 06 July 2010 [18], which are standards for reporting experiments involving animals. Under isoflurane anesthesia, cervical dislocation was used to put rats to sleep. The livers were separated, cleaned, and cooled saline was injected into them. In 10 mM potassium phosphate buffer (pH 7.4) with 1 mM ethylenediamine tetraacetic acid (EDTA) on ice, tissue was chopped and homogenized (1:10 *w*/*v*). The homogenate was centrifuged at 10,000× *g* for 20 min at 4 °C. To obtain the microsomal fraction, the supernatant was further centrifuged at 100,000× *g* for 60 min. The pellet (microsomes) was kept at 20 °C while being suspended in a solution containing 10 mM histidine (pH 7.2), 25% (*v*/*v*) glycerol, 0.1 mM EDTA, and 0.2 mM CaCl_2_. Using bovine serum albumin as the reference material, the Lowry test was used to determine the protein concentration [19].

### 2.2. The Preparation of Plant Extracts

The selection of “Greenfield” trademark tea was based on the product’s variety of species and the tea samples collected from local Kazakh (Almaty) market and they were purchased as bulk products. Additionally, therapeutic plants in the *Asteraceae* family (*A. tomentosum*, *G. kasachstanicum*, *A. schrenkiana*, *A. rutifolia*, *A. cina* and *A. vulgaris*) were collected in the south-eastern region of Kazakhstan (Sarkan and Urzhar districts) in 2022 and were dried according to the standard of drying of medicinal plants and used for investigation. Plant materials were purchased from a local pharmacy and identified by Dr. Alibek Ydyrys and Raushan Kaparbay. Specimens (*A. tomentosum*—No. 2-386, *G. kasachstanicum*—No. 2-330, *A. schrenkiana*—No. 2-36857, *A. rutifolia*—No. 2-3561, *A. cina*—No. 2-36879, *A. vulgaris*—No. 2-35811) were deposited in the Biomedical Research Centre, Al-Farabi Kazakh National University, Almaty, Kazakhstan. A laboratory mill was used to crush and powder the tested plants. As previously mentioned [20], 10 mL of 50% (*v*/*v*) aqueous ethanol was used to extract one g of crushed and powdered dry plant parts for 20 h at room temperature. The mixture was centrifuged at 20,000× *g* for 10 min, and the supernatant was then dried at 37 °C in a rotary evaporator. Stock solutions of the dried extracts (100 mg) were prepared afresh in 50% ethanol prior to use in the investigations.

### 2.3. The Estimation of Total Phenolic and Flavonoid Content

The Folin-Ciocalteu reagent technique [15] was used to determine the total phenolic content of the extracts: Test tubes containing 2.5 mL of 10% Folin-Ciocalteu reagent and 2.0 mL of 2% sodium carbonate solution were filled with 0.5 mL of each extract (1.0 mg/mL), and the tubes were then vigorously shaken. For 15 min, the mixture was incubated at 45 °C with intermittent shaking. PE-5400 UV ultraviolet (Ekroshim LLC, St. Petersburg, Russia) spectrophotometer was used to detect absorbance at 765 nm. To create a calibration curve (range from 0 to 1 mg/mL), gallic acid (purchased from Sigma-Aldrich, Milan, Italy) was used as the standard. In terms of gallic acid equivalents (GAE), the outcomes were reported as g per mg of dry extract.

Rutin was used as the reference in a colorimetric experiment to evaluate the total flavonoid concentrations [16]. Each extract was diluted with 2.0 mL of distilled water and 150 L of 5% sodium nitrate, yielding 0.5 mL (1.0 mg/mL) of each extract. After being combined for 6 min, the final solution was added, then after 15 min at room temperature, added to 150 L of 10% aluminum chloride and 2.0 mL of 1 M sodium hydroxide. The mixes’ absorbances were then gauged at 510 nm. The results were given in mg of dry extract per g of rutin equivalents (RE). Using rutoside solutions in methanol at concentrations between 0.0 and 1.0 mg/mL, the calibration curve was created in the same way.

### 2.4. The Estimation of Lipid Peroxidation in Liver Microsomes

By using the Ohkawa et al. [19] method to measure malondialdehyde content as thiobarbituric acid-reacting substances (TBARS), lipid peroxidation (LPO) was determined. Briefly, for 10 min at 37 °C with continual stirring, liver microsomes were preincubated with vehicle or test agents in a buffer containing 50 mM 2O_4_ KH_2_PO_4_ (pH 7.2) and 145 mM NaCl. After incubating at 95 °C for 60 min, the basal and 0.02 mM Fe^2+^/0.5 mM ascorbate-induced microsomal LPO levels were then assessed in a reaction mixture comprising 0.9 M sodium acetate buffer (pH 3.5), 0.4% sodium dodecyl sulphate (SDS), and 20 mM thiobarbituric acid. The mixture was extracted using n-butanol:pyridine (15:1, *v*/*v*) and centrifuged at 3000× *g* for 5 min after cooling to room temperature. The organic layer was gathered, and the absorbance at 532 nm was determined. As nmol of TBARS per mg of protein, malondialdehyde (MDA) concentration was expressed [20].

### 2.5. The Isolation of Rat Erythrocytes

Rats were humanely put to sleep after having their hearts punctured and blood drawn while under isoflurane anesthesia. White blood cells and plasma were extracted after they were centrifuged at 1000× *g* for 10 min. Erythrocyte pellets were employed right away for osmotic resistance testing after being rinsed twice with a buffer containing 5 mM Na_2_HPO_4_ (pH 7.4) and 150 mM NaCl [21].

### 2.6. The Estimation of the Osmotic Resistance of Erythrocytes

Erythrocytes’ osmotic resistance (ORE) was assessed as previously mentioned [22]. Isolated erythrocytes were treated with a hypotonic solution of NaCl (0.4%) at 37 °C for 20 min, and then they were centrifuged for 10 min at 14,000× *g*. For ten minutes, the vehicle or test agents were preincubated with the separated erythrocytes. Then, at 540 nm, hemoglobin absorbance in the supernatant was determined. The proportion of total hemolysis brought on by 0.1% Na_2_CO_3_ was used to calculate the extent of hemolysis [23].

### 2.7. Statistical Data Analysis

Statistical analysis was performed with the GraphPad Prism 6.0 Program (GraphPad Software, San Diego, CA, USA). Data were statistically analyzed using comparative and descriptive statistical techniques. The results of three separate experiments were provided as the mean standard deviation (SD). Along with the degree of hemolysis, the link between extract concentration and lipid peroxidation was assessed. The nonlinear regression equation was used to generate the Pearson correlation coefficient, *p* ≤ 0.05 and the T-criterion was used to determine whether registered changes in the indices were reliable.

## 3. Results

### 3.1. The Properties of Plant Extracts

Table 1 displays the findings from the investigation of the IC50 for plant extracts, total flavonoids, and polyphenolic component concentration. The order of extracts can be determined by their IC50 value: *A. vulgaris* < *A. cina* < *A. schrenkiana* < *A. tomentosum* < *G. kasachstanicum*, with the IC50 value for the antioxidant properties of *A. rutifolia* being higher than those of other species included in this study (Table 1).

The most notable membrane-stabilizing effects of *A. schrenkiana* and *A. cina* could be determined based on the estimation of membrane-stabilizing qualities that are associated with the IC50 values. Hemolysis was 50% inhibited by an extract of *A. rutifolia* at a concentration of 200 g/mL, but other extracts’ IC50 values could not be determined at the tested concentration levels.

*A. tomentosum*, *A. schrenkiana*, *G. kasachstanicum*, *A. cina*, *A. vulgaris*, and *A. rutifolia* were the plant extracts classified in order of their quantity of phenolic compounds and flavonoids. The information gained was consistent with research on the antioxidant and membrane-stabilizing effects of plant extracts.

### 3.2. The Influence of Herbal Extracts of the Family Asteracea

The *A. cina* extracts had the best membrane-protective properties, as shown in Table 2. Plant ethanolic extracts with concentrations between 0 and 100 g dry substance/mL effectively reduced erythrocyte hemolysis. Erythrocyte fragility accordingly decreased by up to 43.7% ± 2.18% and 89.8% ± 4.47% at a concentration of 100 g/mL. Plantain’s antihemolytic effect is dose-dependent.

At a concentration of 5 g/mL, the phytoextracts of *A. rutifolia*, *A. schrenkiana*, and *A. cina* had no discernible effect on erythrocyte hemolysis; the hemolysis levels were 99.6%, 93.1%, and 92.7%, respectively. However, the protective effects of *A. schrenkiana* and *A. cina* extracts on erythrocyte membranes were reported to be strengthened at doses above 10 g/mL. Notably, *G. kasachstanicum* extracts do not alter hemolysis at the concentrations of 10, 50, or 100 g/mL, but at doses higher than 5 g/mL, erythrocyte membrane resistance was significantly increased (Table 2). An analysis of the research’s findings revealed that all plant extracts had the ability to stabilize membranes and lessen red blood cell hemolysis. The majority of the extracts significantly altered the stability of erythrocyte membranes.

In the course of studying the membrane-protective properties of a number of plants from the *Asteraceae* family, it was revealed that extracts of these plants dose-dependently reduce the level of hemolysis of erythrocytes in vitro, although the effect is not pronounced. When calculating the IC50 index for two species of *Artemisa* L., this figure was approximately 100 μg, while in species such as *A. tomentosum* and *G. kasachstanicum*, the extract concentration having a 50% effect is outside the studied range. *A. schrenkiana* exhibited a significant membrane-stabilizing effect on erythrocyte membranes. The IC50 value was 50 µg. Thus, according to membrane-protective properties, this plant can be ranked as follows: *A. tomentosum*, *G. kasachstanicum*, *A. rutolifolia < A. cina < A. vulgare < A. schrenkiana.*

The erythrocyte suspension was preincubated with herbal extracts before being exposed to a hypoosmotic NaCl solution during the tests. The amount of hemolysis was used to measure osmotic resistance. As shown in Table 2, nearly all of the extracts caused a dose-dependent reduction in cell hemolysis in the concentration range of 0–100 g/mL. These results unequivocally established the antihemolytic action of *A. cina*, *A. schrenkiana*, and *A. vulgaris* extracts, and they also showed that these extracts reduced the level of hemolysis by 44.5%, 42.2%, and 43.2%, respectively. At 100 g/mL concentrations, the extracts of *A. cina*, *A. schrenkiana*, and *A. vulgaris* also exhibited anti-hemolytic activities, lowering the hemolysis of erythrocytes by 43–51%. Similarly, a minor rise in the level of hemolysis of the red blood cells indicates that the *A. tomentosum* extract has hemolytic activities at a low dose (5 μg/mL).

As a result, the extracts from *A. tomentosum*, *G. kasachstanicum*, and *A. vulgaris* exhibit the strongest membrane-stabilizing action among those examined. At a concentration of 5 g/mL, the extracts of these plants demonstrated a significant anti-hemolytic impact, whereas the remaining extracts displayed a similar effect at doses above 10 g/mL.

The findings of a study on the effects of increasing concentrations of water–ethanol extracts of plants in the Asteraceae family on the osmotic resistance of erythrocytes were shown in Figure 1. Out of the six plants represented in the figure, extracts from one plant (*A. vulgaris*) reduced the hemolysis of erythrocytes by up to 60% when the concentration was increased to 0.1 mg (dry matter/mL SI), while extracts from two other plants (*A. cina* and *A. schrenkiana*) reduced hemolysis at this concentration by 50%. While the hemolysis of erythrocytes was reduced by 30–40% when each plant extract (*A. tomentosum* and *G. kasachstanicum*) was used, this had a favorable impact on the resistance of erythrocytes. By increasing the concentration of Artemisia rutolifolia extracts to 0.1 mg of dry matter/mL SI, the hemolysis of erythrocytes was reduced by 10–25%. The findings of the study demonstrated that all of the plants utilized in the project have a membrane-stabilizing impact to varying degrees, which increases erythrocyte resistance and decreases hemolysis.

According to research on the effects of herbal extracts on the peroxidation processes in liver microsomes, *A. cina* has highly expressive antioxidative capabilities (Table 3). In particular, extracts completely suppressed the production of malondialdehyde at doses ranging from 250 to 2500 g per 1 mg protein.

As can be seen from the table, the IC50 index for all of the studied extracts, in contrast to the previous series of experiments, did not exceed the concentration values of the studied range. The most significant antioxidant property was possessed by *A. cina*, while *A. rutolifolia*, *A. vulgare* and *A. schrenkiana* have an antioxidant effect at almost the same level. The properties of marsh weed which inhibit lipid peroxidation processes were almost two times less compared to the extract from Artemisia L. The antioxidant effect of burdock extracts was significantly lower than that of extracts from different types of Artemisia L.

Additionally, all of the plant extracts used in this study have antioxidant qualities that prevent the production of LPO compounds. According to our findings, there are notable antioxidant qualities in the extracts of *A. tomentosum*, *G. kasachstanicum*, *A. schrenkiana*, *A. rutifolia*, *A. cina*, and *A. vulgaris*. At every tested concentration, they were found to virtually entirely prevent the synthesis of LPO products.

The antioxidant properties of the studied plant extracts of the family Asteraceae are listed in Figure 2. As can be seen from the figure, extracts of all studied plants with increasing concentration from 250 μg to 25 00 μg dry substance/mg protein reduce the accumulation of TBA-active products. Extracts of two plants of *G. kasachstanicum* and *A. cina* at a concentration of 250 μg dry substance/mg protein inhibit the process of lipid peroxidation (LPO) by 50%; with an increase in the concentration of extracts, the formation of peroxide products is completely suppressed.

The analysis of the results of the study of plant extracts of the family Asteraceae showed that the following plants have good membrane-stabilizing and antioxidant properties: *A. vulgaris*, *G. kasachstanicum*, *A. schrenkiana*, and *A. vulgaris.*

### 3.3. The Combination Extracts of Plants and Tea Provide Antioxidant and Membrane-Stabilizing Properties

The 50% inhibitory concentration (IC50) is a crucial parameter in pharmacology and toxicology studies. It represents the concentration of a substance (in this case, plant extracts) needed to inhibit a specific biological or biochemical process by 50%. The lower the IC50 value, the more potent the substance is in inhibiting the process. The importance of the combined plant and tea extracts’ 50% inhibitory concentrations (IC50 values) lies in understanding the synergistic or antagonistic effects of these combinations on a specific biological or biochemical activity. If the IC50 values of the combined extracts are significantly lower than the individual extracts alone, it suggests a synergistic effect. This means that the combination of plant and tea extracts is more potent in inhibiting the target activity than each extract on its own. If the IC50 values of the combined extracts are higher than the most potent individual extract, it indicates an antagonistic effect. In this case, the combination is less effective than the most potent individual extract, and the interaction may be interfering with the desired inhibitory activity. Determining the IC50 values for combined extracts helps in optimizing formulations for specific applications. For example, in pharmaceuticals or natural health products, finding the right combination can enhance the therapeutic effects or reduce side effects. Understanding the combined effects of plant and tea extracts can have implications for health and wellness. For instance, if the combination shows enhanced inhibitory effects on oxidative stress, inflammation, or other health-related factors, it could have potential applications in functional foods or dietary supplements.

We compared the plant content of TBA-active products in *A. schrenkiana*, *A. tomentosum*, and *G. kasachstanicum* with black and green tea. We could not study *A. rutifolia*, *A. cina*, and *A. vulgaris* plants because of their bitter taste. As noted, at a concentration of 20 μg/mL, the content of TBA-active products was reduced by almost 60–90%. The results of the study of the extracts’ potentials are presented in Table 4 and Table 5.

The values provided represent the 50% inhibitory concentrations (IC50) for different plant extracts (*A. tomentosum*, *G. kasachstanicum*, *A. schrenkiana*) and types of tea (Green tea, Black tea). The IC50 is a measure used in pharmacology and biochemistry to assess the effectiveness of a substance in inhibiting a specific biological or biochemical function by 50%. In this case, it is likely referring to the inhibitory concentration of these extracts on a certain activity or process. Smaller IC50 values indicate a more potent inhibitory effect, meaning a lower concentration is required for a 50% inhibition. Therefore, in this context *G. kasachstanicum* (3.2 ± 0.8) and *A. schrenkiana* (3.4 ± 0.1) seem to be more potent inhibitors compared to *A. tomentosum* (4.3 ± 0.6). Green tea (9.7 ± 3.1) and black tea (14.8 ± 4.5) have higher IC50 values, suggesting they are less potent inhibitors compared to the plant extracts.

Smaller IC50 values indicate a more potent inhibitory effect, meaning a lower concentration is required for a 50% inhibition. *G. kasachstanicum* in combination with black tea (2.8 ± 2.1) and *A. schrenkiana* in combination with black tea (2.85 ± 0.12) have lower IC50 values, suggesting they are more potent inhibitors in combination with black tea. *A. tomentosum* in combination with black tea (7.8 ± 2.6) has a higher IC50 value, indicating it may be less potent in combination with black tea compared to the other two extracts. *G. kasachstanicum* in combination with green tea (4.3 ± 1.4) and *A. schrenkiana* in combination with green tea (4.4 ± 1.3) have similar IC50 values, suggesting comparable potency in inhibiting the specified activity in combination with green tea. *A. tomentosum* in combination with green tea (5.0 ± 0.7) has a slightly higher IC50 value, indicating it may be slightly less potent in combination with green tea compared to the other two extracts.

No discernible increase in antioxidant and membrane-stabilizing characteristics has been shown when black tea is combined with *A. tomentosum* and *G. kasachstanicum* (Table 4), as opposed to the individual plant extracts. When compared to other plant extracts, the *G. kasachstanicum* extract showed a noticeably stronger antioxidant activity (IC50 = 3.2 ± 0.8). The individual extracts’ IC50 values varied as follows: *G. kasachstanicum* > *A. schrenkiana* > *A. tomentosum*; in combination with black tea, *G. kasachstanicum* > *A. schrenkiana* > *A. tomentosum;* and in combination with green tea, *G. kasachstanicum* > *A. schrenkiana* > *A. tomentosum*. According to this assay, the *G. kasachstanicum* extract combined with black tea was the most active. Furthermore, the extract was found to be more potent than both single extracts and extracts combined with green tea. In individual extracts, *G. kasachstanicum*, and *A. schrenkiana* have lower IC50 values compared to *A. tomentosum*, indicating higher potency. In combinations with both black tea and green tea, *G. kasachstanicum* and *A. schrenkiana* generally exhibit lower IC50 values compared to *A. tomentosum*, suggesting greater potency in combination with tea.

The combination of *G. kasachstanicum* and black tea exhibited a higher membrane-protective effect than either extract used alone. But there was no discernible rise in antioxidant potential. The combination of *A. schrenkiana* with black tea did not result in any notable alterations, and the IC50 remained constant. As a result of a synergistic interaction between the constituents in these extracts, combining *A. tomentosum* extracts with black tea extract, in contrast to the aforementioned plants, increased their antioxidant and membrane-stabilizing activities. Table 5 displays the findings of related investigations using the combinations of green tea and other plant extracts.

The membrane-stabilizing properties of the combination extracts of plants, as indicated by their IC50 values (µg/mL of RBC), provide insights into their potential effects on red blood cells (RBCs) and cellular membranes. Green tea has an intermediate IC50 value, suggesting moderate membrane-stabilizing properties. It may be less potent than *A. tomentosum* and *A. schrenkiana* but more effective than *G. kasachstanicum*. The IC50 value for black tea is greater than 200, which implies that at the concentrations tested, it did not show significant membrane-stabilizing effects. It might not be as effective in this regard compared to the other extracts. *A. tomentosum* and *A. schrenkiana* appear to have similar and relatively strong membrane-stabilizing properties.

*G. kasachstanicum* has a higher IC50 value, indicating potentially weaker membrane-stabilizing effects. Green tea falls in between, with moderate membrane-stabilizing properties. Black tea, at the concentrations tested, did not demonstrate notable membrane-stabilizing effects.

The results suggest that *A. tomentosum* and *A. schrenkiana* could be explored further for their potential benefits in membrane stability, which might have implications for various physiological processes. *G. kasachstanicum*, while still effective, may be less potent in this regard. The differences observed could be attributed to the unique chemical compositions of each plant extract.

It is important to note that these interpretations are based on the given IC50 values, and further studies may be needed to understand the mechanisms and broader implications of these membrane-stabilizing properties.

The results suggest that the combination of *G. kasachstanicum* with black tea and green tea demonstrated the strongest membrane-stabilizing properties among the combinations tested. This could be indicative of a synergistic effect between the components of *G. kasachstanicum* and black tea in enhancing membrane stability. *A. tomentosum* in combination with black tea shows moderate membrane-stabilizing properties. *A. schrenkiana* in combination with black tea has a higher IC50 value, suggesting a potentially weaker effect in stabilizing RBC membranes compared to the other combinations.

Among individual extracts, *G. kasachstanicum* has the highest IC50 value, indicating potentially weaker membrane-stabilizing properties. Green tea and *A. tomentosum* have relatively lower IC50 values, suggesting moderate membrane-stabilizing effects. Combinations with green tea generally show either similar or lower IC50 values, suggesting potential synergistic effects. Combinations with black tea show varied effects, with some combinations having lower IC50 values (potentially enhanced effects) and others having higher values (potentially reduced effects).

These interpretations are based on the IC50 values and assume that lower values indicate stronger membrane-stabilizing effects. The differences observed may be attributed to the unique chemical compositions of each plant extract and how they interact with the components of green tea. Further studies could explore the mechanisms behind these observations.

As mentioned, the combinations of *A. tomentosum* and green tea extracts have antioxidant and antihemolytic qualities comparable to those of the separate plant extracts. When *A. schrenkiana* extracts were combined with green tea extract, similar outcomes were seen, suggesting no appreciable synergistic interaction. On the other hand, combining *G. kasachstanicum* extracts with green tea resulted in an increase in their antioxidant potential and membrane-stabilizing effects, further supporting the idea that the components of these separate plant extracts work in concert.

## 4. Discussion

When analyzing the data, we can conclude that the membrane-protective effect and antioxidant properties of extracts do not manifest themselves to the same extent in the same plant species; this was most likely due to the fact that the bioactive substances in plants have different mechanisms of action on membranes.

Table 2 and Table 3 describe the results of the investigation into how plant extracts affect erythrocyte osmotic resistance and lipid peroxidation processes in the microsomal portion of liver membranes.

When analyzing the data, we can conclude that the membrane-protective effect and antioxidant properties of extracts do not manifest themselves to the same extent in the same plant species; this is most likely due to the fact that the bioactive substances in plants have different mechanisms of action on membranes.

It is widely known that raw plant-based products have beneficial properties for people and contain a wide range of compounds [24,25]. While both green and black teas are produced using the leaves of the plant, there are differences between the processes involved in each tea’s production. Numerous studies have been done on the antioxidant properties of green and black tea [26,27,28].

Different categories of phytochemical substances, which are organic sources of antioxidants, are present in medicinal plants. Tea may play a significant role in the prevention of numerous diseases, according to the high ratio of physiologically active phenolic and polyphenolic components in tea [29,30]. There is proof that tea can lower the risk of cardiovascular illnesses, some types of cancer, and a variety of other chronic diseases, both in vitro and in vivo [31,32]. In an attempt to optimize the antioxidant status of black and green tea and, consequently, enhance their additional protective qualities, recent research has focused on the use of plants with noteworthy antioxidant and membrane-stabilizing characteristics.

Some studies have suggested that extracts from *A. lappa* and *A. tomentosum* may possess membrane-stabilizing properties. These properties are attributed to their bioactive components, which could help maintain the integrity and fluidity of cell membranes, making them more resistant to oxidative and mechanical stressors [33,34]. A systematic analysis of *A. lappa* and potentially other Arctium species could provide a comprehensive understanding of their combined antioxidant and membrane-stabilizing properties. This research may uncover the therapeutic potential of these plants in preventing or alleviating conditions associated with oxidative stress and cellular membrane dysfunction [35].

*G. luteoalbum*, like many other Asteraceae family members, contains bioactive compounds such as flavonoids, polyphenols, and terpenoids, which are known for their antioxidant potential. These compounds can help neutralize harmful free radicals, reducing oxidative stress and protecting cells from damage [36]. While research on the membrane-stabilizing properties of Gnaphalium species is limited compared to more extensively studied plants, the presence of certain bioactive constituents in *G. viscosum* may suggest its potential for membrane protection. These constituents could contribute to the stability and functionality of cell membranes [37].

The biologically active substance, antioxidants, and bioavailability of extracts of *A. vulgaris* was investigated [38,39]. The phytochemical diversity of plants is reflected in their antioxidant and bioavailable capacities. The antioxidant activity of a plant is influenced by the type of phytochemicals it contains. These phytochemicals, which also serve as singlet oxygen scavengers, high-energy radiation absorbers, and plant defense mechanisms include polyphenols, phenolic acids, flavonoids, and alkaloids [39,40]. Flavonoids are present in many therapeutic plants. The ability of flavonoids to modify processes that are dependent on membranes, including the lipid peroxidation brought on by free radicals in membranes, is associated with both their structural characteristics and their ability to interact and integrate into the lipid bilayer [41,42,43].

Other writers have documented the antioxidant properties of a few plant extracts from the Asteraceae family’s *Artemisia* L., such as *A. schrenkiana*, *A. cina*, *A. vulgaris*, and *A. rutifolia*, and attributed their scavenging activity to phenolic and flavonoid contents [44,45,46,47,48].

Lipids serve as the membrane’s building block and are crucial to the stability and structural organization of biological membranes [49]. We have previously demonstrated that every plant extract employed has antioxidant qualities that prevent lipid peroxidation from damaging membranes. There are indications that membrane lipid peroxidation occurs prior to erythrocyte hemolysis [50]. Scientist Alinkina Ekaterina Sergeevna’s tests showed how essential oils affected erythrocyte membranes, and it was found that essential oils and an extract containing phenols, such as eugenol, carvacrol, thymol, polyphenols, gingerols, and gingerones, had the highest activity. Oils containing 1–3% phenol derivatives showed lower N activity. Oils with two double bonds in the cycle and between 10 and 40 percent mono- and sesquiterpenes showed minimal action. The least amount of a- and y-terpinene in essential oils showed the least amount of antiradical activity. For the essential oils of thyme, oregano, and savory, a synergistic interaction between thymol and carvacrol was found. Additionally, it was discovered during the study that oregano essential oils demonstrated the properties of active bio-antioxidants when taken regularly for six months in tiny dosages [51].

It is widely known that the oxidation of the biologically active components causes black tea to lose a significant amount of its beneficial properties during fermentation operations. Green tea has greater advantageous impacts on living things than black tea. On hepatocyte membranes in vitro, both varieties of tea have antioxidant properties that are dosage dependent [52]. When their antioxidant activities (*A. schrenkiana*, *A. tomentosum*, and *G. kasachstanicum*) were compared, it became clear that green tea’s qualities were superior to those of black tea. Additionally, it has been demonstrated that *G. kasachstanicum*, *A. schrenkiana*, and *A. cina* extracts preserve erythrocyte membranes as well as microsomal fractions of hepatocytes.

## 5. Conclusions

The practice of preparing meals, producing beverages, and treating illnesses with plant components is a holdover from prehistoric tribes and civilizations. *A. schrenkiana*, *A. tomentosum*, and *G. kasachstanicum* extracts have particularly strong synergistic effects when combined with common tea beverages, according to the results of a comprehensive investigation into the antioxidant and membrane-stabilizing properties of Kazakhstani medicinal plants from the *Asteracea* family. These extracts may be recommended for potential tea production and further research into the specific compounds responsible for the observed synergy.

When their antioxidant activities were compared, it became clear that green tea’s qualities were superior to those of black tea. The protective effects of *G. kasachstanicum*, *A. schrenkiana*, and *A. cina* extracts on erythrocyte membranes and the microsomal fractions of hepatocytes have also been demonstrated.

## Figures and Tables

**Figure 1 plants-13-00096-f001:**
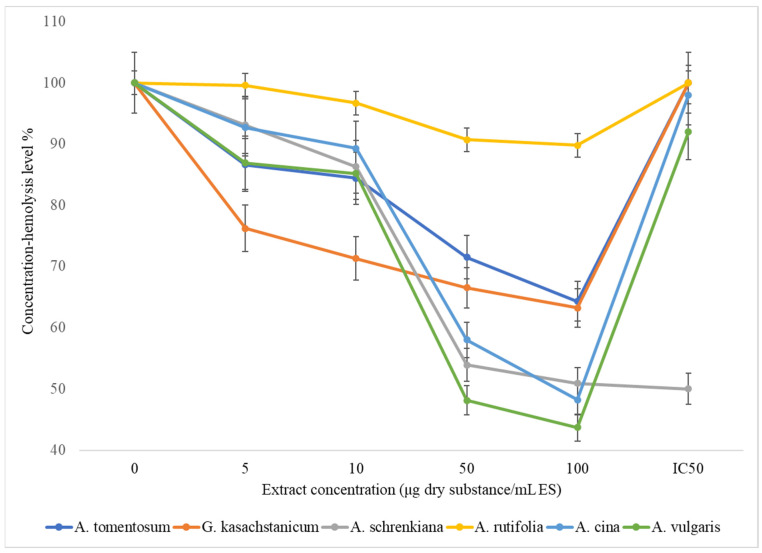
Effects of *Asteraceae* family herbal extracts on erythrocyte membrane osmotic resistance.

**Figure 2 plants-13-00096-f002:**
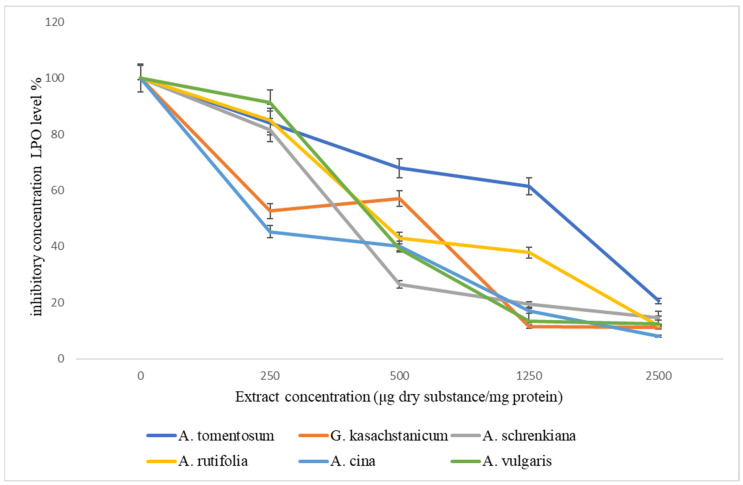
Effect of herb extracts of family *Asteraceae* on the level of lipid peroxidation in the liver microsome.

**Table 1 plants-13-00096-t001:** Lipid peroxidation and membrane-stabilizing properties of several *Asteraceae* family plant extracts (mean ± standard deviation (SD), *n* = 3).

Species	Total Polyphenols (μg GAE/mg)	Total Flavonoids (μg RE/mg)	Lipid Peroxidation IC50 (µg/mg Protein)	Membrane-Stabilizing Properties IC50 (µg/mL of RBC)
*A. tomentosum*	348.4 ± 11.9	214.3 ± 19.2	4.6 ± 0.8	64.2 ± 7.3
*G. kasachstanicum*	312.4 ± 22.3	284.2 ± 38.3	9.7 ± 3.2	183 ± 9.4
*A. schrenkiana*	342.1 ± 2.1 *	265.1 ± 1.2 *	3.4 ± 0.07 **	195.1 ± 7.0 *
*A. rutifolia*	236.4 ± 8.1 *	71.2 ± 6.7 *	4.7 ± 0.6 **	>200
*A. cina*	253.6 ± 8.5 *	131.2 ± 13.1 **	2.6 ± 0.5 ***	183 ± 6.2 *
*A. vulgaris*	240.3 ± 13.7 **	111.1 ± 7.6 *	8.1 ± 2.3 *	64.6 ± 3.6 *

A log dosage concentration–inhibition curve was used to calculate the 50% inhibitory concentration (IC50) values in μg/mL. The concentration of total flavonoids and polyphenols was given as the mean ± SD of studies conducted in triplicate. Values represented as mean ± SE (*n* = 3). *p* value (***): very highly significant; (**): highly significant; (*): significant.

**Table 2 plants-13-00096-t002:** Influence of herbal extracts of the family *Asteraceae* on the osmotic resistance of the erythrocyte membrane. Note: mean ± SD, *n* = 3. The extent of hemolysis was calculated as the percentage of total hemolysis caused by 0.1% Na_2_CO_3_.

Species	Extract Concentration (μg Dry Substance/mL ES)					
	0	5	10	50	100	IC50
*A. tomentosum*	100	86.6 ± 4.33	84.4 ± 4.2	71.5 ± 3.55	64.3 ± 3.21	100 μg <
*G. kasachstanicum*	100	76.2 ± 2.81	71.3 ± 3.4	66.5 ± 2.32	63.2 ± 3.21	100 μg <
*A. schrenkiana*	100	93.1 ± 1.2 *	86.3 ± 3.4 **	53.9 ± 5.7 ***	50.9 ± 0.9 *	50 μg <
*A. rutifolia*	100	99.6 ± 4.92 **	96.7 ± 4.8 ***	90.7 ± 4.51 **	89.8 ± 4.47 ***	100 μg <
*A. cina*	100	92.7 ± 4.63 **	89.3 ± 4.46 *	58.0 ± 2.85 *	48.2 ± 2.4 *	98 μg <
*A. vulgaris*	100	86.9 ± 4.33 **	85.2 ± 4.25 *	48.1 ± 2.4	43.7 ± 2.18 **	92 μg <

The data are expressed as the mean ± SD (*n* = 3). Values represented as mean ± SE (*n* = 3). *p* value (***): very highly significant; (**): highly significant; (*): significant.

**Table 3 plants-13-00096-t003:** Effect of *Asteraceae* family herb extracts on the liver microsome’s degree of lipid peroxidation. Note: *n* = 3, mean ± SD.

Species by Common Name	Extract Concentration (μg Dry Substance/mg Protein)					
	0	250	500	1250	2500	IC50
*A. tomentosum*	100	84.0 ± 4.2	68.0 ± 3.4	61.5 ± 3.1	20.6 ± 1.01	1559 μg
*G. kasachstanicum*	100	52.6 ± 2.96	57.1 ± 2.57	11.5 ± 5.7	11.2 ± 0.3	552 μg
*A. schrenkiana*	100	81.5 ± 3.2 *	26.6 ± 3.7 **	19.4 ± 2.5 **	14.5 ± 2.9 *	392 μg
*A. rutifolia*	100	85.0 ± 4.25 **	43.0 ± 2.1 ***	37.8 ± 1.94 ***	11.8 ± 0.59 ***	444 μg
*A. cina*	100	45.3 ± 2.7 ***	40.0 ± 2.0 ***	17.0 ± 0.85 ***	8.0 ± 0.39 ***	232 μg
*A. vulgaris*	100	91.2 ± 4.5 ***	39.1 ± 1.98 ***	13.3 ± 0.65 ***	12.4 ± 0.62 *	432 μg

In comparison to the control group, the data are presented as the mean ± SD (*n* = 3); * indicates *p* < 0.05; ** indicates *p* < 0.01; and *** indicates *p* < 0.001. *A. tomentosum* rxy = −0.9228, *G. kasachstanicum* rxy = −0.9811, *A. schrenkiana* rxy = −0.9736, *A. rutifolia* rxy = −0.9653, *A. cina* rxy = −0.3612, and *A. vulgaris* rxy = −0.9307.

**Table 4 plants-13-00096-t004:** The combined plant and tea extracts’ 50% inhibitory concentrations (IC50 values) (mean + SD).

No.	Sample	IC50 (µg/mg Protein, Mean + SD)		
		Individual Extract Mean	In Combination with Black Tea	In Combination with Green Tea
1	*A. tomentosum*	4.3 ± 0.6	7.8 ± 2.6	5.0 ± 0.7
2	*G. kasachstanicum*	3.2 ± 0.8	2.8 ± 2.1	4.3 ± 1.4
3	*A. schrenkiana*	3.4 ± 0.1	2.85 ± 0.12	4.4 ± 1.3
4	Green tea	9.7 ± 3.1	-	-
5	Black tea	14.8 ± 4.5	-	-

The mean ± SD (*n* = 3) is used to express the data. A log dosage concentration inhibition curve was used to get the 50% inhibitory concentration (IC50) values in μg/mL.

**Table 5 plants-13-00096-t005:** Membrane-stabilizing properties of combination extracts of plants and tea (mean + SD).

No.	Sample	IC50 (µg/mL of RBC, Mean + SD)		
		Individual Extract Mean	In Combination with Black Tea	In Combination with Green Tea
1	*A. tomentosum*	86.1 ± 7.1	124.0 ± 6.3	89.1 ± 7.2
2	*A. schrenkiana*	195.1 ± 1.5	177.9 ± 9.2	143.22 ± 7.5
3	*G. kasachstanicum*	84.3 ± 4.2	81.0 ± 3.7	76.5 ± 4.5
4	Green tea	114.3 ± 9.5	-	-
5	Black tea	>200	-	-

The mean ± SD (*n* = 3) is used to express the data. A log dosage concentration inhibition curve was used to get the 50% inhibitory concentration (IC50) values in μg/mL.

## Data Availability

Data are contained within the article.

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
