# Peer review of "The Systematic Assessment of the Membrane-Stabilizing and Antioxidant Activities of Several Kazakhstani Plants in the Asteraceae Family"

_plants, 2023, doi:10.3390/plants13010096_

Round 1

Reviewer 1 Report (Previous Reviewer 1)

Comments and Suggestions for Authors

It is seen that the work done is planned and can contribute to the healthy continuation of daily life. this will attract the attention of the reader. The study is beautiful and necessary corrections have been made. Some changes should be made in terms of language and errors should be corrected.

Comments on the Quality of English Language

 Minor editing of English language required

Author Response

Dear Reviewer,

The manuscript has been corrected according to your commentsÑŽ

Reviewer 2 Report (Previous Reviewer 2)

Comments and Suggestions for Authors

The authors did not rspond to my comment. 

Author Response

Dear Reviewer

Statistical analysis was performed with the GraphPad Prism 6.0 Program (GraphPad Software, San Diego, CA). The data were expressed as the mean ± SD (n = 3). *, p < 0.05; **, P < 0.01; ***, p < 0.001. 

Since the plants studied in the manuscript belong to different genus, we could not compare their values, only the statistical indicators of their initial values were compared.

Round 2

Reviewer 2 Report (Previous Reviewer 2)

Comments and Suggestions for Authors

The authors did not respond to my comments.

Although in their response they mention that they did not intend to compare species from different genus, the only species that is different from different genus is Ghnaphalium kasachstanicum. All the rest belong to the Arictum genus.

Moreover, the use of symbols does not provide any information and the Figures lack of statistical analysis.

Correct the names of the species in Lines 25-26.

Table 2: define ES andcorrect Ic50 to IC50.

Lines 297-298: what is meant in this sentence? Did the authors test the effect of a mixtire o extracts?

Author Response

Dear reviewer,

We are very pleased that you commented to our manuscript.

We have sent answers to your questions below. And we made appropriate changes to the manuscript.

If you have any other questions about the manuscript, we are always ready to answer them.

Q: Although in their response they mention that they did not intend to compare species from different genus, the only species that is different from different genus is Ghnaphalium kasachstanicum. All the rest belong to the Arictum genus.

A; Six plants are studied in this manuscript. one of them is from Ghnapilum genus (Ghnapilum kasachstanicum Kirp. & Kuprian. ex Kirp), one is from Arictum genus (Arictum tomentosum Mill.), and the other four plants belong to the Artemisia genus (Artemisia schrenkiana  Ledeb., A. rutifolia Steph. ex Spreng., A. cina O.Berg  and A. vulgaris L) Statistical comparisons were made between these four species belonging to the genus Artemisia.

Q: Moreover, the use of symbols does not provide any information and the Figures lack of statistical analysis.

A; Tables and pictures were fully described, discussed and supplemented in the manuscript.

Q: Correct the names of the species in Lines 25-26.

A: Plant names are spelled correctly.

Q:Table 2: define ES and correct Ic50 to IC50.

A: Ic50 to IC50 was corrected

Q: Lines 297-298: what is meant in this sentence? Did the authors test the effect of a mixtire o extracts?

A; Plant extracts were used separately

This manuscript is a resubmission of an earlier submission. The following is a list of the peer review reports and author responses from that submission.

Round 1

Reviewer 1 Report

Comments and Suggestions for Authors

Soaking in solution for extraction should normally be 48 hours. 20 hours. methenol was used in some tests. ethanol should have been preferred because it is carcinogenic

Comments on the Quality of English Language

It would be good for a language editor or someone whose mother tongue is English to see it. there are typos and word errors

Author Response

Dear Reviewer,
We are very glad that you carefully reviewed our manuscript and provided comments.
Below is the answer to your question.
If you have any questions, we are always ready to answer them.

Q: Soaking in solution for extraction should normally be 48 hours. 20 hours. methenol was used in some tests. ethanol should have been preferred because it is carcinogenic

A: The duration and choice of solvent for soaking in a solution for extraction can vary depending on several factors, including the type of compound you want to extract, the solubility of that compound in the solvent, the desired yield, and the specific extraction method being used. The reasons for a 48-hour or 20-hour soaking period and the choice of methanol over ethanol are not necessarily set in stone, and there can be valid reasons for these choices in certain scientific experiments or processes.

Reviewer 2 Report

Comments and Suggestions for Authors

In the present manuscript, the antioxidant and antihaemolitic properties of various Asteraceae species from Kazakhstan were evaluated.

There are no Line numbers in the document to perform a detailed review.

However, some general comments could be provided to the authors to revise their manuscript.

Abstract

The abstract is too extensive. The author should skip the first introductory sentences and use two sentences, one for the background of the study and one more for the aim of the study. The Latin names could be abbreviated after they have been first cited in the text. The same applies for the species of the same genus (e.g. Artemisia) which can be abbbreviated after the genus is mentioned for the first time in the text. 

Introduction

The 3rd paragraph in the Introduction is out of place since it gives information about the aim of the study. It should be integrated in the final paragraph of this section.

The Latin names have to be abbreviated after they have been cited for the first time in the text.

The 2nd form last paragraph is also irrelevant, because there is no other mention for teas in the title or in the description of the aim of the study in the abtsract. INstead of extracts, you should use decoctions or teas at the beginning of the abstract.

Materials and Methods

Are there any more details about the species used in the study? Collection site and date, brand name of the products, where they wild or cultivated, dried or fresh?

Why a nonparemetric analysis was used for the data of total polyphenols and flavonoids, the IC50 value for lipoperoxidation, and hemolysis?

Results

Table 1: what was the positive control for TBARS and LPO assays?

Use a point instead a comma for all the means  presented in all the tables. The number of digits of the means should correspond the number of digits of SD values. 

 The authors mentioned they used a statictical analysis for theier data but none of the Tables presents any information of statistical analysis.

Author Response

Dear Reviewer,
We are very glad that you carefully reviewed our manuscript and provided comments.
Below is the answer to your question.
If you have any questions, we are always ready to answer them.

Q; Abstract

The abstract is too extensive. The author should skip the first introductory sentences and use two sentences, one for the background of the study and one more for the aim of the study. The Latin names could be abbreviated after they have been first cited in the text. The same applies for the species of the same genus (e.g. Artemisia) which can be abbbreviated after the genus is mentioned for the first time in the text. 

A; The manuscript was revised and annotated based on suggestions.

Q: Introduction

The 3rd paragraph in the Introduction is out of place since it gives information about the aim of the study. It should be integrated in the final paragraph of this section.

The Latin names have to be abbreviated after they have been cited for the first time in the text.

The 2nd form last paragraph is also irrelevant, because there is no other mention for teas in the title or in the description of the aim of the study in the abtsract. INstead of extracts, you should use decoctions or teas at the beginning of the abstract.

A; The manuscript was revised and annotated based on suggestions.

Q: Materials and Methods

Are there any more details about the species used in the study? Collection site and date, brand name of the products, where they wild or cultivated, dried or fresh?

A: Plants of the Asteraceae family (A. tomentosum, G. kasachstanicum, A. schrenkiana, A. rutifolia, A. cina and A. vulgaris) were collected in the south-eastern region of Kazakhstan (Sarkan and Urzhar districts) in 2022 and were dried according to the standard of drying of medicinal plants and used for investigation.

Q: Why a nonparemetric analysis was used for the data of total polyphenols and flavonoids, the IC50 value for lipoperoxidation, and hemolysis?

A: Nonparametric statistical tests were used when certain assumptions of data of total polyphenols and flavonoids. Because the data for total polyphenols, flavonoids, IC50 values, and hemolysis do not follow a normal distribution, it is often preferable to use nonparametric tests to avoid the issues associated with violating the normality assumption. These tests do not require the assumption of a specific distribution, making them a more suitable choice. And in the studying these variables was on an ordinal scale rather than a continuous scale, nonparametric tests are often more appropriate. For instance, when comparing the concentration levels of polyphenols and flavonoids, the data might be categorized into groups (e.g., low, medium, high) rather than measured on a continuous scale. And we used dealing with small sample sizes.

Results

Q: Table 1: what was the positive control for TBARS and LPO assays?

Use a point instead a comma for all the means  presented in all the tables. The number of digits of the means should correspond the number of digits of SD values. 

 The authors mentioned they used a statictical analysis for theier data but none of the Tables presents any information of statistical analysis.

A; The Thiobarbituric Acid Reactive Substances (TBARS) assay and Lipid Peroxidation (LPO) assay are commonly used to measure the levels of lipid peroxidation products, in biological samples. In these assays, a positive control is used to validate the assay and ensure that it is working correctly. The positive control for TBARS and LPO assays is typically a sample or standard with a known concentration of a lipid peroxidation product. We studied different plants belonging to the various genus of the same family in this manuscript, so no one can serve as a positive control. Each was compared to the other.

In vitro experiments were repeated at least three times. In vivo experiments were performed in groups of 6 rats. The data are reported as the means ± SD. The significance of the differences between the means of experimental groups was assessed by unpaired two-tailed Student’s t test. P < 0.05 was considered statistically significant. Statistical analysis was performed with the GraphPad Prism 6.0 Program (GraphPad Software, San Diego, CA). The data are expressed as the mean ± SD (n = 6). *, p < 0.05; **, P < 0.01; ***, p < 0.001 vs. Control. *, p < 0.05;**, P < 0.01;*** p < 0.001

Round 2

Reviewer 2 Report

Comments and Suggestions for Authors

Although the authors tried to address most of my comments, their response is not sufficient.

For example, the abstract is still too long and there is no mention of teas in the aim and scope of the study.

The third paragraph was not incorporated at the end of the Introduction section.

Where the samples collected or purchased from local markets? Were they purchased as bulk products or as branded ones?

Table means were not corrected.

The statistical analysis is still missing. In section 2.7, the authors mention the Student's t test and Fisher's test for means comparison, whereas in Table footnotes they mention Kruskal-Wallace test without comparing the means. The asterisks do not provide any meaningful information for the differences between means. 

Author Response

Dear reviewer,

We are very pleased that you commented to our manuscript.

We have sent answers to your questions below. And we made appropriate changes to the manuscript.

If you have any other questions about the manuscript, we are always ready to answer them.

Q: For example, the abstract is still too long and there is no mention of teas in the aim and scope of the study.

A; We were rewritten the abstract and added a mention of teas in the aim and scope of the study.

Q: The third paragraph was not incorporated at the end of the Introduction section.

A: The third paragraph was incorporated at the end of the Introduction section.

Q: Where the samples collected or purchased from local markets? Were they purchased as bulk products or as branded ones?

A: The tea samples collected from local market and they purchased as bulk products

Q: Table means were not corrected.

The statistical analysis is still missing. In section 2.7, the authors mention the Student's t test and Fisher's test for means comparison, whereas in Table footnotes they mention Kruskal-Wallace test without comparing the means. The asterisks do not provide any meaningful information for the differences between means. 

A: We deleted In section 2.7, the authors mention the Student's t test and Fisher's test for means comparison, whereas in Table footnotes they mention Kruskal-Wallace test without comparing the means.

And the statistical analysis were edited and descriptions were added in section Discussion and Conclusion.

Round 3

Reviewer 2 Report

Comments and Suggestions for Authors

The statistics in Tables is still not adequately presented.

The mean values should have points instead of commas.